# The Role of SOX Transcription Factors in Ageing and Age-Related Diseases

**DOI:** 10.3390/ijms24010851

**Published:** 2023-01-03

**Authors:** Milena Stevanovic, Andrijana Lazic, Marija Schwirtlich, Danijela Stanisavljevic Ninkovic

**Affiliations:** 1Institute of Molecular Genetics and Genetic Engineering, University of Belgrade, Vojvode Stepe 444a, 11042 Belgrade, Serbia; 2Faculty of Biology, University of Belgrade, Studentski trg 16, 11158 Belgrade, Serbia; 3Serbian Academy of Sciences and Arts, Knez Mihailova 35, 11000 Belgrade, Serbia

**Keywords:** SOX, ageing, age-related diseases, stem cell exhaustion, neural stem cells

## Abstract

The quest for eternal youth and immortality is as old as humankind. Ageing is an inevitable physiological process accompanied by many functional declines that are driving factors for age-related diseases. Stem cell exhaustion is one of the major hallmarks of ageing. The SOX transcription factors play well-known roles in self-renewal and differentiation of both embryonic and adult stem cells. As a consequence of ageing, the repertoire of adult stem cells present in various organs steadily declines, and their dysfunction/death could lead to reduced regenerative potential and development of age-related diseases. Thus, restoring the function of aged stem cells, inducing their regenerative potential, and slowing down the ageing process are critical for improving the health span and, consequently, the lifespan of humans. Reprograming factors, including SOX family members, emerge as crucial players in rejuvenation. This review focuses on the roles of SOX transcription factors in stem cell exhaustion and age-related diseases, including neurodegenerative diseases, visual deterioration, chronic obstructive pulmonary disease, osteoporosis, and age-related cancers. A better understanding of the molecular mechanisms of ageing and the roles of SOX transcription factors in this process could open new avenues for developing novel strategies that will delay ageing and prevent age-related diseases.

## 1. Introduction

Ageing is an inevitable process that affects all living beings. Although hard to define, ageing is represented as a set of functions that begin to decline after sexual maturity and progress with age [1,2]. Human life span has greatly increased in the past few decades, and about 20% of the global population is expected to be older than 60 years of age by 2050 [3]. However, the extension of health span, the period of healthy life free of chronic diseases or disabilities of ageing, is yet to be achieved [4].

Ageing is accompanied with many functional declines at molecular, cellular, tissue, and organismal levels [5]. These alterations include but are not restricted to genome instability, telomere attrition, epigenetic alterations, loss of proteostasis, deregulated nutrient sensing, stem cell exhaustion, mitochondria dysfunction, cellular senescence and altered intercellular communications, which are all known as nine major hallmarks of ageing (Figure 1) [2,6]. Due to these alterations, ageing is considered to be a driving factor for development of age-related diseases such as neurodegenerative diseases, heart failure, chronic obstructive pulmonary disease (COPD), chronic kidney disease, osteoarthritis, stroke, and type 2 diabetes mellitus. Age-related diseases also include some common cancers such as breast, prostate, leukaemia, and colorectal cancer which incidence is increased with ageing. Improved life expectancy will unavoidably increase age-related pathological conditions that will bring economic and psychological burden on patients, their families, and society as a whole [7]. However, underlying age-related mechanisms undeniably leading to age-related phenotypes are still unclear. 

Since ageing research has the potential to reveal new solutions for the treatment of age-related diseases, there is a great interest in studies focused on molecular mechanisms underlying this complex process. 

Many molecular events associated with ageing originate from global chromatin remodelling, leading to the changes in the nuclear architecture and altered gene expression associated with ageing [8]. Studies on the age-related transcriptomic changes in the tissues across the whole body represent valuable information for a better understanding of ageing [9]. Therefore, it is not surprising that there is an increase in studies of the ageing transcriptome [10]. These studies reveal the gene expression alteration in many ageing cells and organs [10]. In addition, age-gene expression signatures display tissue-specific pattern [9]. Frenk et al. described six gene expression hallmarks of cellular ageing: down-regulation of genes encoding mitochondrial proteins; down-regulation of the protein synthesis machinery; dysregulation of the immune system genes; reduced growth factor signalling; constitutive responses to stress and DNA damage; dysregulation of gene expression and mRNA processing [10]. Identifying altered gene expression patterns in the course of ageing is also important for recognising new biomarkers for detecting the effect of different environmental factors on the rate of ageing [10].

Transcription factors (TFs) are important regulators of gene expression for nearly every aspect of the cell function [5]. Besides gene expression alteration, changes in TFs activity are also substantial during ageing [2]. In addition, the age-related decline is regulated by a broad network of signalling pathways, many of which involve downstream TFs [11]. Together with chromatin modification, TFs can be regarded as an important hub linking many ageing hallmarks [5].

SOX (Sry-related HMG box) proteins constitute a large family of evolutionary conserved TFs comprising at least 20 SOX family members in mammals [12]. The genes encoding SOX proteins have been designated as *Sox* or *SOX* in animals and humans, respectively. SOX proteins display properties of both classical TFs and architectural components of chromatin [13], and they exert regulatory functions to activate or repress gene transcription through specific interactions with their partner factor(s) [14].

SOX TFs are a component of a regulatory network and, together with other TFs, signalling pathways, epigenetic modifiers, and microRNAs govern diverse cellular processes during development, such as the maintenance of stem cell pluripotency, cell proliferation, cell fate decisions, germ layer formation and the terminal differentiation of cells into tissues and organs [15]. However, the roles of SOX TFs are not limited to the development as they also affect cell survival, regeneration and death, and control homeostasis in the adult tissues [16,17,18].

In this review, we mostly focused on the roles of SOX TFs in stem cell exhaustion and age-related diseases that are correlated with this ageing hallmark, including neurodegenerative diseases, visual deterioration, COPD, osteoporosis and age-related cancers. Additionally, we have given a brief overview of possible correlation between these transcription factors and other major hallmarks of ageing such as telomere attrition and cellular senescence. Finally, we summarize current perspectives regarding anti-ageing therapies including anti-ageing drugs that effect *SOX* genes expression. All in all, a better understanding of the roles of *SOX* genes in ageing could advance the development and application of specific pharmaceutical approaches in preventing or delaying age-related diseases.

## 2. Alternation of SOX Genes Expression during Ageing

It has been recently shown that some members of *SOX* gene family display altered expression with ageing in various cells and tissues, in both mice and humans. Expression of *Sox2*/*SOX2*, *Sox4*/*SOX4*, *Sox10*/*SOX10* and *Sox17*/*SOX17* are decreased during ageing, and the expression of *Sox15*/*SOX15* is increased, while the alternation of *Sox9*/*SOX9* expression in ageing is tissue-specific (Table 1). The reduced level of *SOX2* expression in peripheral blood mononuclear cells observed in elderly individuals suggests that the level of *SOX2* expression could be considered as a biomarker of ageing assessable through a non-invasive blood test [19]. Considering results obtained upon microarray data of differentially expressed genes in skin fibroblasts of elderly and young individuals *SOX9* was suggested as biomarker of skin ageing [20].

One of the main epigenetic phenomena shared between aged mouse and human was ageing-associated DNA methylation changes found at TF-binding sites. Interestingly, DNA methylation alteration in both species was commonly found in HMG domains of several SOX TFs [21].

**Table 1 ijms-24-00851-t001:** Altered expression of *Sox*/*SOX* genes during ageing.

Gene	Expression during Ageing	Cells/Tissue	Species	References
** *Sox1* **	Decreased	NPCs	Mice	[22]
***Sox2*/*SOX2***	Decreased	NPCs, peripheral blood mononuclear cells brain, lung, heart, kidney, spleen and liver	Mice, Human	[19,23,24]
** *SOX4* **	Decreased	Luminal epithelial cells in mammary tissue	Human	[25]
***Sox9*/*SOX9***	Increased	Tendon	Human	[26]
	Decreased	Articular chondrocytes growth plates chondrocytesarticular cartilage, skin	Mice, Human	[27,28,29]
***Sox10*/*SOX10***	Decreased	Cochlear lateral wall	Mice, Human	[30]
***Sox17*/*SOX17***	Decreased	Skeletal muscle	Human	[31]
** *SOX15* **	Increased	Luminal epithelial cells in mammary tissue	Human	[25]

## 3. SOX Transcription Factors and Life Span

Partial Sox2+ stem cell exhaustion in mice resulted in premature ageing, where mice display pronounced spinal kyphosis, hair greying, and accumulation of senescent cells in the tissues, all well-known signs of ageing [32]. A mouse model with down-regulated expression of *Sox4* in the most tissues was viable and fertile, but these animals developed a wide range of premature ageing-associated pathologies with significantly reduced life span [33]. Furthermore, a decreased expression of *SOX4* was detected in fibroblasts of patients with Hutchinson–Gilford progeria syndrome, a rare genetic disease with widespread phenotypic features resembling premature ageing [34].

## 4. SOX Transcription Factors and Telomere Attrition

It has been observed that decreased expression of *SOX4* affected the shortening of telomere length of peripheral blood cells, a well-known hallmark of both cellular senescence and organismal ageing [33]. Additionally, another member of the *SOXC* group, *SOX11*, was linked to telomere length. It was observed that with increased maternal age at birth, telomere length in their children’s fully differentiated T-cells (CD45RA+CD57+) is also increased [35]. This positive association between maternal age and telomere length is mediated by decreased methylation of CpG islands in the promoter of *SOX11*. How methylation of *SOX11* is implicated in telomere length variability still needs to be elucidated [35].

## 5. Senescence and SOX Transcription Factors

Senescence, one of the hallmarks of ageing, represents a cellular response to various types of stress signals triggered by intrinsic and/or extrinsic stimuli [36,37,38]. Cellular senescence, through the paracrine release of IL-6 and other soluble factors, strongly favours cellular reprogramming by Oct4, Sox2, Klf4 and c- Myc in non-senescent cells [39]. In ageing tissues, the expression of *SOX2* inversely correlates with the expression of p16^INK4a^, a well-known cell cycle inhibitor and established senescence biomarker [19]. Cho et al. have shown that SOX2-induced autophagy enhanced cellular senescence by up-regulating tumour suppressors or senescence factors, including p16, p21 and phosphorylated p53 (Ser15) [40]. Deletion of *Sox2* in cultured osteoblast cell lines leads to a senescence-like phenotype, while its overexpression prevents differentiation [41]. Adult Sox2+ stem cell exhaustion in mice results in cellular senescence and premature ageing, indicating that adult stem cell exhaustion can lead to the cellular senescence induction and premature ageing [32].

Ink4a- and Arf-deficient mice showed that SOX5 exerts tumour-specific effects through cellular senescence activation [42]. The authors show that over-expression of *Sox5* gene leads to an increase in the number of big flat cells and senescence associated beta galactosidase positive cells, both characteristics of senescence. Further, they analysed the mechanism of *Sox5*-induced senescence and showed that SOX5 did not affect the p16^Ink4a^ pathway, but the effect was observed on activation of p27^kip1^. They suggest that acute cellular response mediated by SOX5 depends on p19^Arf^ but not p16^Ink4a^ [42]. It was shown that transgenic mice with over-expression of *Sox9* could avoid senescence, and cells exhibited normal fibroblast morphology compared to the controls [43]. They also showed that the expression of Ink4b and DcR2, established senescence markers, was inversely correlated with *Sox9* expression [43]. These results suggest that SOX9 is an essential player in the regulation of cellular senescence [43].

Advances in understanding molecular mechanisms underlying senescence that are at least partly associated with SOX TFs would provide the opportunity to modulate this complex phenotype during ageing and in various pathophysiological contexts.

## 6. SOX and Stemness-Related Ageing

Most mammalian adult tissues contain resident tissue-specific stem cells responsible for homeostatic tissue maintenance and regenerative responsiveness to injury. Stem cells are undifferentiated and unique by their ability to produce differentiated daughter cells and retain their stem cell identity by self-renewal [44]. Due to their capacity to self-renew and differentiate, adult stem cells are often considered immortal reservoirs of youth [45]. They serve as a cellular backup of various organs, continuously renewing them by supplying new daughter cells with the set of cytokines and growth factors required for the regeneration and repairing of tissues/organs and keeping them integrated and functional throughout their life span [46]. During ageing, stem cells show multiple signs of functional decline compromising their self-renewal and proliferation. These age-associated progressively deteriorating changes deprive the organ-specific niches/microenvironment of functional growth and molecules needed for mutually concerted functioning between stem cells and their respective organs [47]. Menendez and Vazquez-Martinargue have suggested that “we are as old as our adult stem cells are” because endogenous stem cells become damaged as we age [48]. Accordingly, stem cell exhaustion manifested through their quantitative and/or qualitative decline has been recognized as one of the drivers of ageing [2]. Resident stem cells maintain the balance between cell loss and cell replacement until such equilibrium starts to be progressively reduced, and newborn cells cannot compensate for the ones that died [49,50]. Stem cell exhaustion, together with altered intercellular communications, represents the end result of other hallmarks of ageing and is ultimately responsible for the functional decline associated with ageing [2]. Molecular ageing of stem cells includes diminished homeostatic control, impaired stem cell response, senescence, apoptosis, aberrant differentiation, somatic mutations, altered functions and failed self-renewal (Figure 2) [51,52]. These changes are accompanied by altered expression of genes in ageing stem cells [53].

The age-related decline in stem cell number and functions is suggested to occur in virtually all tissues and organs maintained by adult stem cells, such as bone, muscle and the forebrain [54,55,56]. Moreover, the loss of tissue homeostasis and regenerative capacity during life can lead to the development of various age-related pathologies, as discussed in Section 7.1, Section 7.2, Section 7.3, Section 7.4 and Section 7.5.

Numerous studies have reported the roles of SOX TFs in preserving stem cell characteristics, playing a part in the regulatory network required to establish embryonic stem cells (ESCs) and to maintain their pluripotent and proliferative state. In particular, SOX2, through cooperative interaction with NANOG and OCT4, drives pluripotent-specific expression of the numerous genes and at the same time, SOX2, OCT4, and NANOG regulate their own expression via positive-feedback loops in ESCs [57]. This core transcriptional circuit consisting of SOX2, OCT4, and NANOG orchestrates the maintenance of stem cell self-renewal and pluripotency, while the fine-tuning of *SOX* genes expression levels controls the balance between cell stemness and differentiation [57,58].

Many data implicate that some *Sox*/*SOX* genes are indispensable for maintaining many types of adult stem cells [59,60,61,62], as summarized in Table 2. Thus, a systematic fate mapping analysis of *Sox2* expression, performed by Arnold and co-workers, demonstrated that immature Sox2+ cells are present in numerous adult tissues, including testes, forestomach, glandular stomach, trachea, anus, cervix, oesophagus, lens and dental epithelium and that they give rise to all mature cell types within these tissues [59]. Similarly, *Sox9* is expressed in several endoderm-derived and ectoderm-derived tissues, including stem/progenitor cells in the adult liver, exocrine pancreas and intestine that represent a source for continuous production of hepatocytes, acinar cells and enterocytes, respectively, under both homeostatic and certain injury conditions [60]. Interestingly, *Sox8* and *Sox15* mark muscle satellite cells and their individual over-expression in the myoblast cell line prevents myotubes formation [61,62]. SOXB1 members are crucial for self-renewal and differentiation of neural progenitor cells (NPCs) both in neurogenic zones of the adult brain [63,64,65] as well as in the retina [66]. Numerous in vitro and in vivo studies indicate that the level of *Sox*/*SOX* genes expression must be tightly controlled for proper neural development [63,67]. The exhaustion of these SOX+ neural progenitors lead to different age-related neurodegenerative disorders, which will be discussed in more detail later in this review (Section 7.1).

## 7. *SOX* Genes and Age-Related Diseases

Ageing is characterized by physiological loss of tissue homeostasis, commonly associated with a progressive and extensive decline in the physical and cognitive performance of the whole organism. In addition to the loss of stem cell turnover, ageing is also caused by the decrease in the overall function of differentiated cells due to the several cell-intrinsic and environmental factors, which may lead to numerous age-related diseases [97,98,99,100]. For instance, age-related hematopoietic stem cells (HSCs) lead to diminished production of adaptive immune cells with an increased incidence of anaemia and myeloid malignancies in aged organisms [101]. As mentioned above, SOX TFs are implicated in the stem and progenitor cells maintenance within numerous adult tissues (Table 2) and alteration in their expression is correlated with various age-related diseases, including neurodegenerative diseases, vision deterioration, chronic obstructive pulmonary disease, osteoporosis and cancers as discussed in more details within this section. It is most likely that the alternation in *Sox*/*SOX* genes expression affects the self-renewal capacitance of resident stem cells, causing the reduction of stem cells pool or inducing their aberrant differentiation leading to pathological outcome (Figure 3).

### 7.1. Neurodegenerative Diseases

Neurodegenerative diseases represent age-related conditions with multifactorial aetiologies caused by progressive loss of selectively vulnerable populations of neurons and, consequently, the loss of cognitive and physical functions [106]. Multiple studies from recent years suggested deregulated neurogenesis as a pathological condition, even in the early phases of the disease, in several neurodegenerative disorders that display symptoms related to hippocampal and sensory dysfunction, including Alzheimer’s, Parkinson’s and Huntington’s diseases [107,108].

In the adult mammalian brain neurogenesis, continuous generation of new neurons from neural stem cells (NSCs) occurs mostly in the two main neurogenic niches: the lateral subventricular zone (SVZ) and subgranular zone (SGZ) of the hippocampal dental gyrus [23].

Numerous studies showed that the self-renewal and/or differentiation capacities of NSCs and NPCs in neurogenic regions of the adult brain are regulated by several *SOX* genes: *SOX2*, *SOX21*, *SOX9*, *SOX4* and *SOX11* [63,64,65,88,109,110]. Furthermore, a recent study in the hippocampus showed the critical role of SOX5 and SOX6 in the transition of NSCs from quiescence to activation [89].

In the course of adult neurogenesis *Sox2*, *Sox3*, *Sox1*, *Sox9* and *Sox21* are highly expressed in undifferentiated cells (NSCs and/or NPCs), and their expression is diminished gradually during the differentiation of mature neurons and glial cells (reviewed in [16]). Additionally, *SOXB* genes show dynamic changes in the expression pattern in the course of in vitro neural differentiation [111]. Thus, the decline in the neurogenic potential of NSCs during ageing could depend on the proper expression of these genes. Comparative studies, both in rodents and humans, showed that the expression of *Sox2*/*SOX2* significantly decreases in the hippocampus during ageing [19,23]. Bharathi Hattiangady et al. showed that a decline in neurogenesis during ageing in rats is not attributable to the decrease in the total number of SOX2 positive cells in the hippocampus but to the increased quiescence, lengthening of cell cycle times of NSCs with ageing, which is correlated with the reduced number of SOX2+ proliferative (Ki67+) cells [112]. However, one comparative study demonstrated a reduction of SOX2 positive cells during ageing in primates’, but not in mice’s hippocampus [113]. A direct correlation between a dramatic decrease in the adult neurogenesis due to stem cell exhaustion during ageing and decreased number of SOX2+ progenitor cells is also detected in another brain region—the hypothalamic arcuate nucleus, the region responsible for the regulation of homeostasis and ageing [24]. SOX1 is also an essential transcription factor in maintaining the NPCs pool [69]. In particular, it has been shown that *Sox1* is expressed in the hippocampus in an activated stem/progenitor cell population, which can give rise to new neurons and astrocytes [69]. The number of SOX1^+^ NSCs declines with age, although the stability at the molecular level was preserved [69]. In fact, microarray analysis showed slight differences in the expression of only a few genes when SOX1+ radial astrocytes derived from young and old mice were compared [69]. Kupiers et al. also described the pool of proliferative (BrdU+)/Sox1+ cells in the hippocampus which might represent a reservoir of quiescent early NPCs capable of reactivation and differentiation in neurons [22]. However, they showed only a moderate decrease in the number of these Sox1^+^ cells in the SGZ from early to middle adulthood in mice [22]. Discrepancies seen in different studies of *Sox1* and *Sox2* expression in the brain during ageing may come from different experimental conditions used in these studies, such as age and gender of animals or markers used to characterize neurogenesis.

Even though other *SOX* genes are known to have an essential role in neurogenesis, their expression and function in the ageing brain are yet to be revealed. Having in mind the critical role of *SOX* genes in the maintenance of NSCs and NPCs, we might suggest that alternations of adult neurogenesis in the aged brain, increased quiescence of NSCs and lengthened cell cycle of NPCs are directly correlated with decreased SOX1 and SOX2 expression detected in the hippocampus during ageing (Figure 4). These changes in *SOX* genes expression could also contribute to diminished number of newborn neurons, representing a risk for neurodegenerative disease development. 

The correlation between the decline of SOX2+ or SOX1+ in NPCs within specific neurogenic niches of the adult brain and different age-related neurodegenerative disorders has been reported. Accordingly, it has been recently demonstrated that depletion in SOX2 positive NSCs in the hippocampus of Alzheimer’s patients correlated with the severity of the disease or the patient’s cognitive capacity [98]. In our previous work on rodents, we analysed the expression of selected members of the SOXB group in the SGZ of the hippocampus of 5xFAD mice, representing a transgenic model of Alzheimer’s disease (AD). Immunohistochemical analysis revealed a significant decrease in the number of cells expressing SOX1, SOX2, and SOX21 within the SGZ of 2 months old 5xFAD mice compared to their non-transgenic counterparts [102]. However, further studies are needed to clarify the involvement of *SOX* genes in the onset and progression of AD.

We previously presented several *SOX* genes (*SOX2*, *SOX6*, *SOX9*, *SOX4*) as potential miRNAs targets in neurodegenerative diseases, including AD, Parkinson’s and Huntington’s diseases. We also suggested that modulation of these *SOX* genes expression by miRNAs might be considered as potential tool for future treatment of neurodegenerative diseases [114].

### 7.2. Visual Deterioration

Numerous degenerative changes in anatomy, physiology and neurological system of the eye emerge with ageing. Visual deterioration is mainly associated with the degeneration of the retina and decline in Müller glia cell activity [115,116]. In the adult retina, Müller cells are the principal cells responsible for the maintenance of retinal homeostasis, structure and metabolism [117]. Several studies demonstrated the critical role of SOX2 in NPCs proliferation and differentiation in the retina both during development and adulthood [66,118]. A functional study suggested that SOX2 promotes Müller, and amacrine cells differentiation partly through the induction of *Pax6* and by facilitating the cell cycle exit of the retinal progenitor cells [119]. Another study demonstrated that SOX2 has a key role in maintaining Müller cells shape and quiescence state to prevent their premature re-entering of the cell cycle and subsequent depletion through cell divisions [17]. During adult life, *Sox2* remains expressed in differentiated cells, ganglion cells (cholinergic), cholinergic subset of amacrine cells, and Müller cells [103,120]. A comparative study in elderly/ageing mice retina demonstrated a decline in the number of Sox2-positive Müller, amacrine and ganglion cells [103,120]. The critical role of *Sox2* in the age-associated decline in retinal cell function and visual activity was recently demonstrated in mice. Studying the impact of *Sox2* haploinsufficiency on the activity of Müller cells and vision loss with age, Moreno-Cugnon with colleagues demonstrated that this TF is required for the maintenance of the transmission of visual information from cones and rods and that retinal cell ageing and age-related vision loss is a consequence of the decline of *Sox2* expression [103].

### 7.3. Chronic Obstructive Pulmonary Disease (COPD)

The lungs consist of various stem/progenitor cells in each separate tissue that makes up the lung as a whole, representing a critical reservoir for maintaining tissue homeostasis and responding to injury [99]. Due to these pools of stem/progenitor cells, the lungs maintain remarkable phenotypic plasticity [121]. As a consequence of ageing, lung stem/progenitor cell regenerative capacity is diminished, leading to lung failure [99]. COPD is a lung disease that develops gradually due to the age-related loss of pulmonary function [122]. Clara cells are the peripheral airways progenitor cells involved in regeneration and immune responses following injury. These cells are vital for pulmonary homeostasis, and it was suggested that they play a critical role in COPD progression [123]. In the postnatal lungs, SOX2 is required to maintain and differentiate Clara cells into respiratory epithelium cells [104]. Loss of *Sox2* expression in respiratory epithelium led to progressive loss of differentiated ciliated, Clara and goblet cells that originate from Clara progenitor cells [104]. By inhibiting the TGF-ß1/Smad3 signalling pathway, SOX17 stimulates respiratory epithelial progenitor cell identity and lineage specificity in the mature lung [121]. These results indicate the important roles of *SOX* genes in the proliferation and differentiation of pulmonary progenitor cells which is important for maintenance of lung homeostasis. Contribution of the involvement of *SOX* genes in COPD is based on the fact that *SOX5* was identified as a candidate gene for COPD. The expression of *SOX5* was reduced, and significantly correlated with lung function, as observed in the lung tissue of some subjects with COPD [124].

### 7.4. Osteoporosis

Bone is a highly active tissue that displays continuous self-regeneration throughout adulthood to maintain structural integrity in the process known as bone remodelling [100]. As a consequence of ageing, there is a decrease in bone mineral density leading to osteoporosis, another disease that represents a major health problem among the elderly worldwide [125]. The bone remodelling process is orchestrated by different cell types originating from mesenchymal or hematopoietic precursors [100]. In bone, the process of osteogenesis is driven by a sequential cascade of biological processes initiated by the recruitment of mesenchymal stem cells (MSCs) to bone remodelling sites [100].

*SOX5* was found to be up regulated in human MSCs isolated from bone marrow samples of postmenopausal osteoporosis patients, and it was recognized as a therapeutic target through regulation of the KLF4 signalling pathway in the treatment of this osteoporosis in females [105]. Up-regulation of *SOX5* decreased the alkaline phosphatase activity and the expression of osteoblast markers, including *Collagen I*, *Runx2* and *Osterix*, which further resulted in inhibition of osteogenic differentiation of human MSCs in females [105].

SOX11 was also found to be implicated in osteoporosis. miR-204-5p-*SOX11* axis was proposed to have a role in ageing osteoblasts through downstream regulation of BMPR1A/Runx2 signalling involved in osteoporosis. In addition, it was suggested that SOX11 is essential for maintaining joint homeostasis and the bone healing process [126]. The authors also suggest that the decrease in SOX11 expression can result in the cell cycle arrest of osteoblasts, leading to clinical observations of increased bone loss among the elderly [126]. Apart from the role in stem/progenitor cells, it was suggested that polymorphisms in *SOX4* and *SOX6* genes are related to osteoporosis [127,128,129,130]. In addition, reduced expression of *SOX4* was found in patients with postmenopausal osteoporosis [131] while decreased Sox4 expression in mice resulted in reduced bone mineral density [132].

### 7.5. Cancer

Ageing is an important driver of malignant transformation [133]. Incidence of breast, prostate, acute myeloid leukaemia (AML) and colorectal cancer increases with age [134,135,136,137,138]. This is in correlation with the fact that in progressively ageing population, the number of older patients with cancer is increasing significantly [139]. Older cancer patients often face difficulties and many side effects of anticancer treatments. Deregulated expression of almost every member of the SOX family is connected to at least one of the cancer types, and several good reviews describe their multiple roles in the development and progression of cancer [140,141,142]. Besides deregulation, many *SOX* genes are amplified in human cancers [140,142]. Depending on the cellular context, *SOX* genes can act as oncogene and/or tumour suppressors [140]. They are involved in different tumour-associated processes, including proliferation, survival, migration, invasion, and metastasis [143,144,145]. It is particularly important to point out the roles of SOX TFs in regulating maintenance of cancer stem cells (CSCs). Age-associated effects increase the resistance to cellular senescence and programmed cell death and may ultimately lead to the transformation of stem cells into CSCs. These cells display stem cell-like properties, and it is suggested that they have a central role in the carcinogenesis process [133]. SOX2 is recognized as a good candidate for a biomarker of CSCs in the prostate cancer, where these cells are important for the recurrence of this cancer [146]. It was also suggested that SOX2 might be associated with quiescent features of CSCs [147,148]. In addition, it was demonstrated that SOX2 plays a crucial role in colorectal CSCs and that SOX2 may play a role in the tumour progression and recurrence in colorectal cancer [146]. SOX17 TF is known for transforming foetal hematopoietic cells into adult hematopoietic cells [149]. *Sox17* expression in HSCs decreased during late foetal development and was no longer detectable two weeks after birth. Ectopic expression of *Sox17* in the adult HSCs was sufficient to increase the self-renewal potential and the reconstituting potential of adult hematopoietic cells [96]. It was also shown that reduction of *SOX17* expression in AML patients is associated with poor prognosis [150].

## 8. Effects of Anti-Ageing Drugs on *SOX* Genes Expression

The effects of several anti-ageing drugs on stem cells have been tested, including dasatinib, imatinib, metformin, rapamycin and quercetin, and we emphasize their influence on *SOX* genes expression.

Dasatinib is a second-generation tyrosine kinase inhibitor (TKI) for chronic, blastic, or accelerated phase chronic myeloid leukaemia (CML) patients [151]. It has been shown that dasatinib promotes the activation of quiescent HSCs in mice [152].

Imatinib-mesylate (IM) enhances the maintenance of CML stem cell potential [153]. Bono et al. have evaluated the impact of IM on the maintenance of stem cell phenotype of K562 cells and expression of stem cells marker in K562 cells under glucose shortage. They have shown that NANOG and SOX2 protein expression is enhanced by IM, underlying its effects on stem cell potential.

Targeting NSCs response to the transient pharmaceutical intervention with imatinib in the middle-aged brain was sufficient to transiently boost NSCs activation rates back to younger levels. [154]. By analysing transcriptomic changes associated with NSCs deeper quiescence, authors have shown changes in the expression of several genes during neurogenesis, including *Sox11* and *Sox30*.

Metformin is the most frequently prescribed antidiabetic drug [155]. Numerous studies have shown that metformin modulates many physiological and pathological processes ranging from ageing and cancer to fracture healing by targeting stem cells [156,157]. Moreover, metformin exerts anti-ageing effects and has beneficial effects against many other diseases, including cardiovascular and autoimmune diseases [48].

Metformin significantly decreased Cdk5 and increased *Sox6* during cell differentiation and promoted neuronal differentiation via crosstalk between Cdk5 and SOX6 in neuroblastoma cells [158]. Treatment with metformin significantly increased neurite length, the number of cells with neurite, and the expression of neuronal differentiation markers in neuroblastoma cells.

Quercetin reduces the expression of *SOX2* in human pluripotent embryonal carcinoma NT2/D1 cells and lung cancer stem cells, while the expression of this gene was significantly increased with quercetin treatment of human dental pulp stem cells [159,160,161].

Rapamycin significantly decreased *Sox1* expression in mESCs and *SOX2* expression in hESCs [162,163]. The effect of rapamycin on SOX9 was context dependent, while no effect on the expression of *Sox10* in oligodendrocyte progenitors and *Sox2* in mESCs was detected [163,164,165].

Senolytics, compounds that selectively clear senescent cells, show variable effects on SOX9 expression. While Fisetin decreased the expression of SOX9 in human osteoarthritis chondrocytes, combining dasatinib and quercetin increased the number of SOX9^+^ chondrogenic progenitor cells in an in vivo intermittent hydrostatic pressure model in the posttraumatic osteoarthritis rats [166,167].

The presented data suggest that treatment with anti-ageing drugs, besides having effects on various stem cells, also affects *SOX* genes expression.

## 9. Current Anti-Ageing Therapies and Challenges

Short-lived model organisms such as yeast, worms, flies, killifish, mice, and rats have been widely used for studying various genetic, dietary, and pharmacological interventions that increase life span [7,168,169]. The achieved breakthroughs and their impacts on health and disease hold promise that human ageing could be delayed leading to unprecedented health benefits [3]. The overview of anti-ageing therapies targeting ageing hallmarks is presented in Figure 5. They include treatment with suppressors of genomic instability, epigenetic drugs, drugs targeting telomere attrition, drug-based control of protein homeostasis, drugs targeting deregulated nutrient sensing, drugs targeting stem cell exhaustion, mitochondria-based therapies, and drugs targeting cellular senescence and altered intercellular communications.

Considering the crucial roles of SOX TFs in the maintenance of stemness, stem cells self-renewal and differentiation, particular research focus in this review has been made on strategies targeting aged stem cells. Some research has reached preclinical trials (ClinicalTrials.gov) and is likely to have clinical relevance in the future [7,170].

**Figure 5 ijms-24-00851-f005:**
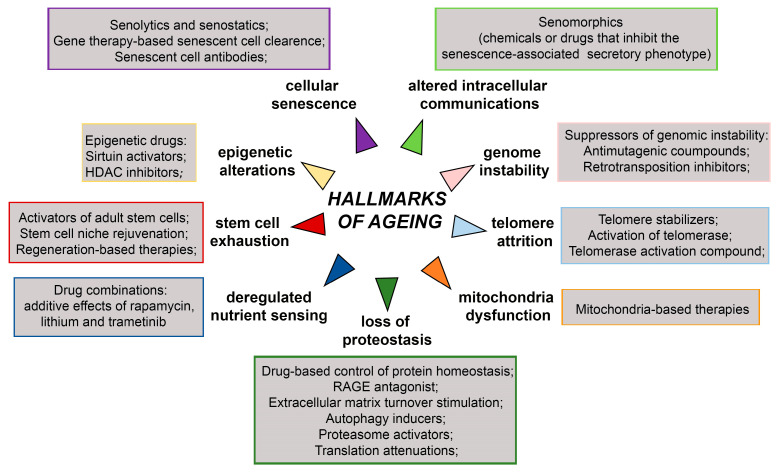
Schematic representation of anti-ageing therapies targeting hallmarks of ageing. HDAC—Hystone deacetilase, RAGE- receptor for advanced glycation end products. The scheme is based on the previously reported publications [7,171,172].

Various therapeutics strategies have been attempted to rejuvenate dysfunctional aged stem cells [74,75]; cellular reprogramming is undoubtedly the most intriguing among them. Revolutionary work by Takahashi and Yamanaka showed that over-expression of four transcription factors (Oct3/4, Sox2, Klf4 and c-Myc, now referred to as the “Yamanaka factors” or “OSKM” factors) rearranges the epigenetic landscape and ultimately reverts somatic cells to a pluripotent state. These reprogrammed somatic cells, with properties similar to ESCs, are named induced pluripotent stem cells (iPSCs) [76,77]. In addition to bringing cells back to a state of immaturity, somatic cell reprogramming was shown by a number of independent studies to erase many signs of cellular ageing, re-establishing a youthful cellular state in cells originating from old individuals [78,79,80,81]. In particular, it was shown that telomeres of iPSCs are longer than in the parent differentiated cells and are comparable in length to telomeres of control ESCs [80]. In addition, other cellular hallmarks of ageing, including disrupted nuclear envelope integrity and mitochondrial function, are rejuvenated by reprogramming [81].

As reviewed by Buganim Y et al., the reprogramming process can roughly be divided into two phases: an initial stochastic phase followed by a hierarchical, deterministic activation of certain genes, with *Sox2* appearing to be a central component of the loop [173]. Indeed, *Sox2* represents one gene of a group of late pluripotency initiating factors crucial for activating the core pluripotency circuitry. Once activated, the endogenous pluripotency proteins OCT4, SOX2 and NANOG occupy their target genes and maintain the iPSCs state in the absence of the exogenous OSKM factors [173]. Besides SOX2, *Sox15* is expressed in mouse ESCs and associated with OCT3/4 binds similar DNA sequences to SOX2 but with weaker affinity [174]. Functional redundancy is commonly present among *SOX* genes. Thus, it was demonstrated that SOX15 could functionally substitute SOX2 in the self-renewal of mouse ESCs [175]. Interestingly, other *Sox* genes have also been used as reprogramming factors. For instance, *Sox1* generated iPSCs with similar efficiency as *Sox2* [176]. Additionally, *Sox3*, *Sox15* and *Sox18* genes could yield iPSCs, although less efficiently than *Sox2* [176]. Another approach is focused on in vivo reprogramming at the organismal level. However, while pulsing of reprogramming factors has been shown to generate short-term beneficial effects, long-term expression of these factors causes teratoma formation [177]. In an attempt to achieve the benefits of reprogramming while minimizing the risk of cancer, a novel approach of partial in vivo reprogramming was introduced. Transient expression of OSKM in the transgenic model of progeria (LAKI-4F mice) was shown to alleviate hallmarks of ageing [178]. The authors also showed that short-term pulsing of the OSKM factors in these mice was also associated with significant improvement in the overall health and lifespan.

The latest research has shown that partial reprogramming by expression of reprogramming OSKM factors for short periods is safe and effective in preventing age-related physiological changes leading to an extended life span of a premature ageing mouse model [179]. The same authors revealed that long-term partial reprogramming led to rejuvenating effects in different tissues, as well as at the organismal level, while the extent of the beneficial effects depended on the duration of the treatment. They also concluded that longer-term partial reprogramming regimens are more effective in delaying ageing phenotypes than short-term reprogramming [179].

Another direction for reprogramming is focused transcription-mediated reprogramming of adult, and even aged cells, to adult tissue-specific stem cells able to compensate for stem cell populations that have become depleted or functionally impaired [170].

An additional avenue of research is focused on transdifferentiation which directly converts differentiated cells of one type into another differentiated cells type. Unlike dedifferentiation, transdifferentiation involves transformation occurring between distinct cell types. Transdifferentiation is based on factors leading to either loss of old or induction of new identity [169]. An appropriate selection of TFs is essential for successful transdifferentiation. Implementing transdifferentiation is still far from clinical application and applying a more natural route of fate-change without reverting to pluripotency could remove many of the potential side effects of such therapy.

Many of the currently discussed strategies are still in their infancy and facing many challenges; thus, further research is required for their translation into clinical practice. For instance, the effects of stem cell-based therapies should be analysed systematically, and effects on every stem cell compartment should be studied; whether the improvements in specific stem cell compartments will cause toxicity elsewhere in the body or long-term stem cell exhaustion, senescence, or dysfunction should be properly examined; appropriate longitudinal studies are required to elucidate the long term and potential adverse side effects [170].

New class of drugs known as senolytics, that selectively induce apoptosis in senescent cells, has attracted attention as a possible treatment option for aging and aging-related diseases. For instance, the senescence-based therapeutic strategy has been shown as a promising approach for osteoporosis treatment. [55]. Recent findings indicate that senescent astrocytes and the accompanying senescence-associated secretory phenotype are correlated with different neurodegenerative disorders [180]. Accordingly, the elimination or reversal of cell senescence by genetic manipulations or pharmacological approaches may exert beneficial effects on neurodegenerative brains [181,182,183]. 

Pharmacological interventions based on the application of selected drugs and small molecules are the most clinically promising strategies for rejuvenating aged stem cells. Many small molecule strategies are hampered by inadequate biodistribution and toxicity from interactions with off-target cells. However, some reach clinical trials for treating ageing and age-related disease. [170]. While partial reprogramming strategies based on transient mRNA expression holds promise, additional research is needed to better understand the full long-term effects of partial reprogramming.

Anti-ageing drugs, also known as geroprotectors, are pharmacological agents that decrease the rate of ageing and extend lifespan [184]. Although anti-ageing treatments have been shown to extend the lifespan of model organisms, there are still no clinically proven human geroprotectors. Initiating some anti-ageing treatments may require early intervention and might not be efficient if subjects are already old, while other treatment forms may show promising results in the elderly but cause unwanted, harmful side effects in healthy, young subjects [185].

Advancements in high-throughput technology provide novel opportunities for an integrated multi-omics approach for better understanding complex molecular mechanisms of ageing. As the gold standard in different fields of biological sciences, the multi-omics approach based on genomics, transcriptomics, proteomics, metabolomics, integromics, microbiomics and systems biology can provide a more comprehensive overview and make a significant contribution to the identification of new candidate biomarkers for ageing, novel targets for anti-ageing interventions and reveal the interactions among ageing molecules from a multidimensional perspective [186]. Combined experimental data from multiple omics methods with computational models will provide a holistic view of the ageing landscape [187]. Undoubtedly, integromics and systems biology will pave the way for the development of personalized ant-ageing treatment.

There are still many challenges that should be addressed to improve the efficiency of anti-ageing strategies. Novel drug delivery vehicles may increase efficacy and decrease toxicity by ensuring that the drug is delivered only to the target cell of interest. Anderson et al. have shown that siRNA-loaded nanoparticles functionalization of nanostructured scaffolds enables controlled delivery to localized regions [188]. Barcoded nanoparticles enable the development of powerful biodistribution screens capable of screening upwards of thousands of nanoparticle subtypes per experiment [189]. Advanced bioinformatics techniques and integromics developed nanoparticles with moieties capable of specific delivery to cellular targets open the avenue for developing nanoparticles capable of targeted delivery toward resident stem cells [170]. The same authors have proposed that progress in stem cell rejuvenation will be made by utilizing a combination of various strategies such as combining drugs, longitudinal studies, secretome isolation and amplification, nanoparticle delivery, single cell characterization, large-scale drug screening, and artificial intelligence and machine learning guided targeting, discovery, and optimization of drugs.

## 10. Concluding Remarks

The quest for eternal youth and immortality is as old as humankind. In the last three decades, tremendous efforts have been made towards understanding the complex multistep process of aging with the main goal to delay or even revert the ageing. However, ageing is an inevitable physiological process accompanied by many functional declines at molecular, cellular, tissue, and organismal levels. Although the human life span has dramatically increased in the past few decades, the extension of the health span is not keeping pace with the increasing life expectancy. Therefore, additional research that will enable the extension of the health span is needed.

Here, we provide the evidence that SOX TFs are associated with many hallmarks of ageing. We have shown that SOX family members display altered expression during the ageing in various cells and tissues, that partial Sox2+ stem cell exhaustion in mice resulted in premature ageing, and that decreased expression of *SOX4* has an effect on the shortening of telomere length. Furthermore, we emphasize the effects of anti-ageing drugs on stem cells and *SOX* genes expression. We also underline that those molecular mechanisms underlying senescence are, at least in part, associated with SOX TFs. 

The most important data presented in this review point to the roles of SOX TFs in preserving stem cell characteristics, and playing a part of regulatory network required to establish ESCs and maintain their pluripotent and proliferative state. SOX2, OCT4 and NANOG comprise the core transcriptional circuit that orchestrates the maintenance of stem cell self-renewal and pluripotency, while the fine-tuning of *SOX* genes expression levels controls the balance between cell stemness and differentiation.

It is widely accepted that restoring the function of aged stem cells is a critical for improving the health span and consequently the lifespan of humans. Accordingly, one of the essential goals is to restore stem cell functions, induce their regenerative potential and slow down the ageing process. Hence, SOX2 emerge as a crucial player in the process of rejuvenation. It is interesting to point out that other SOX TFs have also been used as reprogramming factors, although less efficiently than SOX2 [176]. Lastly, our particular focus has been made on the role of SOX TFs in stem cell exhaustion and age-related diseases, including neurodegenerative diseases, COPD, osteoporosis, and age-related cancers.

The presented data point to SOX TFs as emerging factors that play essential roles during ageing. A better understanding of the molecular mechanisms of ageing and the roles of SOX TFs could open new avenues for developing novel strategies that will delay ageing and prevent age-related diseases.

## Figures and Tables

**Figure 1 ijms-24-00851-f001:**
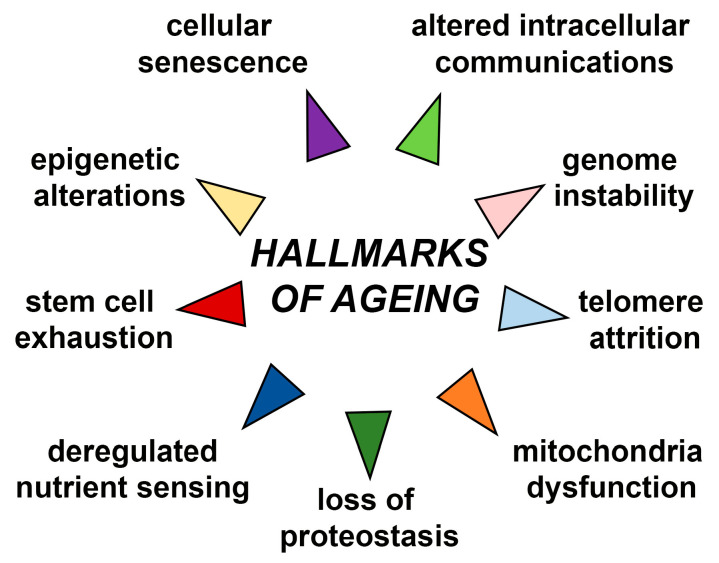
Schematic representation of nine major hallmarks of ageing. The scheme is based on the previously reported publication [2].

**Figure 2 ijms-24-00851-f002:**
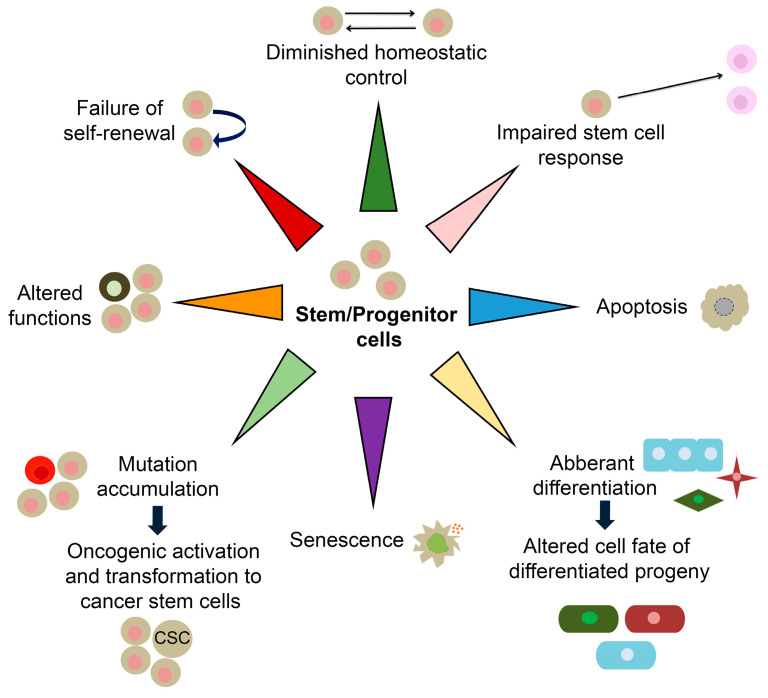
Schematic representation of molecular features of aged stem/progenitor cells. CSC-cancer stem cell. The scheme is based on the previously reported publications [51,52].

**Figure 3 ijms-24-00851-f003:**
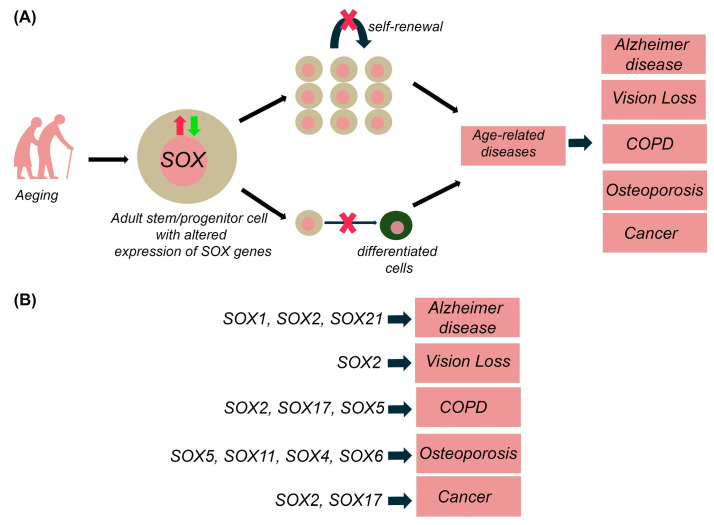
Role of *SOX* genes in stem cell exhaustion and age-related diseases. (**A**) Altered expression of *SOX* genes in adult stem/progenitor cells lead to decrease of self-renewal capacity and inhibition of differentiation of these cells, leading to age-related diseases—Alzheimer diseases, Vision loss, Chronic obstructive pulmonary disease (COPD), osteoporosis, and age-related cancers. (**B**) Alteration in *SOX* genes expression is correlated with various age-related diseases, including neurodegenerative diseases, vision loss, (COPD), osteoporosis, and cancers. The scheme is based on the previously reported publications [96,98,102,103,104,105].

**Figure 4 ijms-24-00851-f004:**
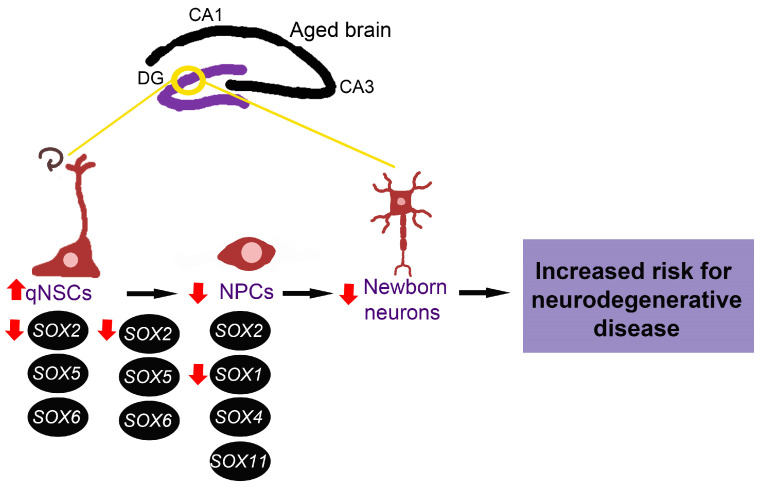
Possible involvement of *SOX* genes in impaired adult neurogenesis during ageing. Alternations of adult neurogenesis that are detected with ageing, increased quiescence of NSCs and lengthened cell cycle of NPCs, could be directly correlated with decreased SOX1 and SOX2 expression detected in the hippocampus during ageing. The expression and function of SOX4, SOX5, SOX6, SOX11 in ageing brain (hippocampus) are yet to be revealed. qNSCs—quiescent neural stem cells, NPCs—neural progenitor cells, DG—dentate gyrus, CA1—cornu ammonis 1, CA3—cornu ammonis 3. The scheme is based on the previously reported publications [16,19,22,23,89].

**Table 2 ijms-24-00851-t002:** *SOX* genes expression in adult stem/progenitor cells.

Group		Gene	Expression in Adult Stem/Progenitor Cells	References
*SOXA*		*SRY*	N/A	
*SOXB*	*SOXB1*	*SOX1*	NPCs in SGZ and SVZ	[63,68,69]
			Radial glial cells in cerebellum	
			(Bergmann cell glia population)	[70,71,72]
		*SOX2*	NSCs and NPCs in SGZ and SVZ	[64,65,73,74,75,76]
			Radial glial cells in cerebellum	[70,71,72]
			(Bergmann cell glia population)	
			Progenitor cells in retina	[66,77]
			Lens stem cells	[59]
			Ganglion stem cells	[59]
			Tracheal epithelial cells	[78]
			Progenitor cells of pituitary gland	[79]
			Tongue epithelium progenitor cells	[80]
			Hear follicle precursor cells	[81,82]
			Olfactory epithelium stem cells	[83,84]
			Glandular stomach stem cells	[59]
			Spermatogonia stem cells	[59]
			Stem cells in cervix	[59]
			Stem cells in esophagus	[59]
			Stem cells in anus	[59]
		*SOX3*	NSCs, NPCs in SGZ, NPCs in SVZ	[85]
			Spermatogonial stem/progenitor cell population)	[86,87]
	*SOXB2*	*SOX14*	N/A	
		*SOX21*	NSCs and NPCs in SGZ	[88]
*SOXC*	*SOX4*		N/A	
	*SOX11*		N/A	
	*SOX12*			
	*(Previously named as SOX22)*		N/A	
*SOXD*	*SOX5*		NSCs and NPCs in SGZ	[89]
	*SOX6*		NSCs and NPCs in SGZ	[89]
	*SOX13*		N/A	
*SOXE*	*SOX8*		N/A	
	*SOX9*		NSCs and NPCs in SVZ	[90]
			Radial glial cells in the cerebellum	[70,71,72]
			(Bergmann cell glia population)	
			Cartilage/tendon stem cells	[60]
			Liver progenitor cells	[60]
			Biliary tree stem cells	[91]
			Exocrine pancreas progenitor cells	[60]
			Intestine progenitor cells	[60,92]
			Hair follicle stem cells	[93]
			Retinal	
			Mammary stem cells	[94]
	*SOX10*		Mammary epithelial cells	[95]
*SOXF*	*SOX7*		N/A	
	*SOX17*		Biliary tree stem cells	[91]
			Hematopoietic stem cells	[96]
	*SOX18*		N/A	
*SOXG*	*SOX15*		N/A	
*SOXH*	*SOX30*		N/A	

## Data Availability

Not applicable.

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
