# Peer review of "The Role of SOX Transcription Factors in Ageing and Age-Related Diseases"

_ijms, 2023, doi:10.3390/ijms24010851_

Round 1

Reviewer 1 Report

The review titled, " The role of SOX transcription factors in ageing and age-related  diseases", covers current known knowledge in the literature in regards to SOX genes and age related diseases as elaborately described in this review to a good extent.

The structure of the review is coherent and easy to follow, however, I would suggest that the authors work on the figures more diligently. For instance, figure 1 looks  unaesthetic and substandard to look at. 

Also, please check for grammatical and typing errors (for eg. line 271 seems to having a typing error). 

I conclusion, with following minor changes, I support the work put in by the authors and recommend for publishing.

Author Response

Our response to Reviewer 1

Reviewer 1:

The review titled, " The role of SOX transcription factors in ageing and age-related diseases", covers current known knowledge in the literature in regards to SOX genes and age related diseases as elaborately described in this review to a good extent.

The structure of the review is coherent and easy to follow, however, I would suggest that the authors work on the figures more diligently. For instance, figure 1 looks unaesthetic and substandard to look at.

Also, please check for grammatical and typing errors (for eg. line 271 seems to having a typing error).

I conclusion, with following minor changes, I support the work put in by the authors and recommend for publishing.

Our response:

We thank the Reviewer for the suggestions.

Accordingly, we modified the Figure 1 and the Figure 5. Additionally, we introduced some improvements in Figures 2 and 4. New versions of figures are included in the revised Manuscript.

All typographical errors are corrected throughout the Manuscript and it was edited for proper English language, grammar, punctuation, spelling and overall style.

Reviewer 2 Report

The review aims to cover the complexity associated with aging and age-related disorders in the context of the SOX transcription factors. The article is well-written and focuses on the roles of SOX transcription factors in stem cell exhaustion and age-related diseases, including neurodegenerative diseases, visual deterioration, chronic obstructive pulmonary disease, osteoporosis, and age-related cancers. The authors have included the latest and relevant information to justify the significance of the article.

I do believe that this manuscript would benefit from reducing the information and giving a more clear-cut design of the scope of the review. 

However, there are a few typographical and grammatical errors that need to be corrected before publication, like, “Line 120: ….could be considered a biomarker” should be as “could be considered as a biomarker”. Articles, especially ‘the’, tend to either be used in abundance or are lacking. The text should be edited so that it flows better for a reader. In general, the manuscript would greatly benefit from grammatical editing.

Author Response

Our response to Reviewer 2

Reviewer 2:

The review aims to cover the complexity associated with aging and age-related disorders in the context of the SOX transcription factors. The article is well-written and focuses on the roles of SOX transcription factors in stem cell exhaustion and age-related diseases, including neurodegenerative diseases, visual deterioration, chronic obstructive pulmonary disease, osteoporosis, and age-related cancers. The authors have included the latest and relevant information to justify the significance of the article.

I do believe that this manuscript would benefit from reducing the information and giving a more clear-cut design of the scope of the review.

However, there are a few typographical and grammatical errors that need to be corrected before publication, like, “Line 120: ….could be considered a biomarker” should be as “could be considered as a biomarker”. Articles, especially ‘the’, tend to either be used in abundance or are lacking. The text should be edited so that it flows better for a reader. In general, the manuscript would greatly benefit from grammatical editing.

Our response:

We thank the Reviewer for the comments.

All typographical errors are corrected and Manuscript was edited for proper English language, grammar, punctuation, spelling, and overall style.

We appreciate the comment about reduction and accordingly we made some changes in the text in effort to reduce the size but remain accuracy and the scope of the Manuscript. All the changes made to the Manuscript are in Track Changes.
